# Association of *CYP2R1* and *VDR* Polymorphisms with Metabolic Syndrome Components in Non-Diabetic Brazilian Adolescents

**DOI:** 10.3390/nu14214612

**Published:** 2022-11-02

**Authors:** Eduarda Pontes dos Santos Araújo, Severina Carla Vieira da Cunha Lima, Ony Araújo Galdino, Ricardo Fernando Arrais, Karla Simone Costa de Souza, Adriana Augusto de Rezende

**Affiliations:** 1Postgraduate Program in Health Sciences, Center for Health Sciences, Federal University of Rio Grande do Norte, Natal 59012-300, RN, Brazil; 2Department of Nutrition, Center for Health Sciences, Federal University of Rio Grande do Norte, Natal 59056-000, RN, Brazil; 3Department of Clinical and Toxicological Analyses, Center for Health Sciences, Federal University of Rio Grande do Norte, Natal 59012-300, RN, Brazil; 4Department of Pediatrics, Center for Health Sciences, Federal University of Rio Grande do Norte, Natal 59012-300, RN, Brazil

**Keywords:** metabolic syndrome, vitamin D, genetic polymorphisms, *VDR*, *CYP2R1*

## Abstract

Associations between vitamin D deficiency and metabolic syndrome (MS) have been reported; however, the underlying biological mechanisms remain controversial. The aim of this study was to investigate the associations of *CYP2R1* and *VDR* variants with MS and MS components in non-diabetic Brazilian adolescents. This cross-sectional study included 174 adolescents who were classified as overweight/obese. Three *CYP2R1* variants and four *VDR* variants were identified by allelic discrimination. The *CYP2R1* polymorphisms, rs12794714 (GG genotype) (odds ratio [OR] = 3.54, 95% confidence interval [CI] = 1.24–10.14, *p* = 0.023) and rs10741657 (recessive model—GG genotype) (OR = 3.90, 95%CI = 1.18–12.92, *p* = 0.026) were significantly associated with an increased risk of MS and hyperglycemia, respectively. The AG + GG genotype (dominant model) of the rs2060793 *CYP2R1* polymorphism was associated with hyperglycemia protection (OR = 0.28, 95%CI = 0.08–0.92, *p* = 0.037). Furthermore, the CC genotype (recessive model) of the rs7975232 *VDR* polymorphism was significantly associated with a risk of hypertension (OR = 5.91, 95%CI = 1.91–18.32, *p* = 0.002). In conclusion, the *CYP2R1* rs12794714 polymorphism could be considered a possible new molecular marker for predicting the risk of MS; *CYP2R1* rs10741657 polymorphism and *VDR* rs7975232 polymorphism are associated with an increased risk of diabetes and hypertension in adolescents with overweight/obesity.

## 1. Introduction

Obesity is an important risk factor for the development of several metabolic abnormalities that can compromise health and quality of life in the long term; among these abnormalities, abdominal obesity, arterial hypertension, dyslipidemia, and glycemic alterations are considered components of metabolic syndrome (MS) [1,2]. MS is associated with a 1.5- to 2-fold increased risk of cardiovascular disease and other causes of mortality in adult and pediatric groups [3].

Data from large observational studies suggest that obesity is associated with an increased risk of hypovitaminosis D [4,5]. Vitamin D is a multifaceted hormone that exerts pleiotropic effects on metabolism, immunity, cell proliferation, and cell differentiation owing to its anti-inflammatory, antiatherogenic, cardioprotective, and neuroprotective properties; additionally, it plays a role in adipogenesis and insulin homeostasis [6,7,8]. However, the causality of the relationship between obesity and hypovitaminosis D remains poorly understood [9,10]. In addition to its relationship with obesity, previous studies have identified an inverse association between vitamin D concentrations and risk factors for MS, such as dyslipidemia, insulin resistance, and arterial hypertension, with serious implications for cardiometabolic risk and morbidity [11,12,13]. 

In this context, it is known that the biological functions of vitamin D are mediated by several genes involved in its metabolism and synthesis [14]. Furthermore, there is a possible association of the polymorphisms of genes that encode the enzymes responsible for vitamin D synthesis with different outcomes, such as the risk of obesity and MS [15]. Among the main genes involved in the bioactivation of vitamin D, *CYP2R1* is responsible for the hydroxylation of vitamin D3 to 25-hydroxyvitamin D3 (25(OH)D). This process initially occurs when the 25-hydroxylase enzyme (encoded by the *CYP2R1* gene) converts vitamin D (inactive precursor derived from sunlight exposure or dietary intake) to 25(OH)D (circulating form) in the liver. Human *CYP2R1* belongs to the cytochrome p450 family and is located at position 11p15.2, encoding an enzyme with 501 amino acids. A mutation in this gene is naturally associated with a deficiency in the active form of vitamin D [16,17]. Despite the limited number of studies on *CYP2R1*, there is evidence suggesting that *CYP2R1* variants affect body mass index (BMI) regardless of vitamin D concentrations. Furthermore, no studies in the literature have investigated the association of MS with variants of this gene [18,19].

Unlike in the case of *CYP2R1*, variants of the vitamin D receptor (*VDR*) gene are the main markers that have been investigated in previous studies on the association of vitamin D with metabolic outcomes. The genomic actions of 1.25 dihydroxyvitamin D3 (1,25(OH)2D3) are mediated by the *VDR*, which belongs to the family of steroid receptors and is expressed in various body tissues (adipocytes, kidneys, pancreas, and immune system) [20]. Along with the *VDR*, 1,25(OH)2D3 regulates the transcription of target genes by heterodimerization with the retinoid X receptor (RXR). The *VDR*-*RXR* complex can translocate to the cell nucleus and interact with deoxyribonucleic acid (DNA) sequence elements or vitamin D response elements (VDREs), which are found in the promoter regions of target genes. Thus, genetic alterations in the *VDR* can lead to important defects in gene activation, cell proliferation and differentiation, calcium homeostasis, and other related biological mechanisms [21]. 

*VDR* polymorphisms, such as rs7975232 (*Apa*I), rs1544410 (*Bsm*I), rs2228570 (*Fok*I), and rs731236 (*Taq*I), which are the most commonly studied, have been associated with MS and its components, including anthropometric and biochemical parameters, in different populations [14,22]. However, corresponding information is not only scarce but also inconclusive. Recently, Jin et al. determined that *VDR* polymorphisms, including *Apa*I, *Bsm*I, *Fok*I, and *Taq*I, were not associated with the risk of MS; however, the *ApaI* variant was associated with hypertriglyceridemia, and the *BsmI* and *TaqI* variants were associated with high-density lipoprotein cholesterol (HDL-c) in adults [23]. By contrast, in Chinese children, the *FokI* polymorphism of the *VDR* was associated with a higher risk of MS [24].

In this study, we aimed to investigate the association of *CYP2R1* polymorphisms (rs10741657, rs2060793, rs12794714) and *VDR* polymorphisms (rs2228570, rs1544410, rs7975232, rs731236) with MS, MS components (waist circumference, fasting blood glucose and triglycerides, blood pressure, HDL-c), and vitamin D serum levels in Brazilian non-diabetic adolescents. 

## 2. Materials and Methods

### 2.1. Study Design and Participants

This was a cross-sectional study involving adolescents aged 10–19 years of both sexes who were overweight/obese. All participants were recruited from the Pediatric Endocrinology Outpatient Clinic of Onofre Lopes University Hospital (HUOL/Natal-RN/Brazil) between September 2017 and March 2020. The determination of the sample size of 174 adolescents was based on a power of 62% required to reveal a difference between the serum levels of vitamin D corresponding to the three groups of alleles of each analyzed polymorphism, on assuming a standard deviation of 0.5 (3.5 ng/mL) and adopting a significance level of 0.017. This sample size made it possible to obtain population estimates of the proportion of individuals with each polymorphism, with a maximum error of 6.2% for a confidence level (CI) of 95%.

The study participants were classified according to the anthropometric nutritional status using the World Health Organization (WHO) BMI/age curves according to sex; z-scores between +1 and +2 indicated overweight, z-scores between +2 and +3 indicated obesity, and z-scores ≥ +3 indicated severe obesity [25]. Then, the adolescents were subdivided into groups with MS (*n* = 48) and without MS (*n* = 126), according to the criteria proposed by the International Diabetes Federation (IDF) [26]. 

The exclusion criteria were the presence of genetic syndromes associated with obesity or other chronic diseases; pregnancy and lactation; use of vitamin D supplementation; use of drugs to treat insulin resistance or type 2 *Diabetes mellitus*; acute or chronic liver, kidney, thyroid dysfunction, heart failure; cancer; or other conditions that alter vitamin D metabolism. 

This study was approved by the Research Ethics Committee of the University Hospital Onofre Lopes (CEP/HUOL/UFRN; CAAE 56763716.7.0000.5292) and was performed in accordance with the principles of the Declaration of Helsinki. Written consent was obtained from all participants and their legal guardians.

### 2.2. Collection of Data and Blood Samples

All participants underwent weight, height, waist circumference (WC), and blood pressure (BP) measurements. All measurements were performed by trained researchers. Weight and height were measured using a digital scale and stadiometer (Omron Health Care, Kyoto, Japan), respectively. WC was measured with an inelastic tape placed at the midpoint between the last rib and iliac crest, with the patient standing up. The WC cutoff point was defined as ≥the 90th percentile. BMI z-scores based on age were calculated according to WHO guidelines [25]. BP was measured after a 5-min rest period using an automatic sphygmomanometer (Omron Health Care, Kyoto, Japan) according to the collection and classification guidelines established by the Brazilian Guidelines on Arterial Hypertension [27]. 

Blood samples were collected after fasting for 12–14 h. The collected blood serum was used to obtain the concentrations of 25(OH)D, fasting glucose, total cholesterol and its fractions, and triglycerides. Biochemical determinations were performed using Wiener kits, according to the methodology described by the manufacturer, and a CMD-800 biochemical analyzer (Wiener Laboratories, Rosario, Argentina). The 25(OH)D analyses were performed using the automated electrochemiluminescence immunoassay kit manufactured by Roche Diagnostics GmbH, COBAS Series 6000 Modular Analyzer (Mannheim, Germany). A diagnosis of vitamin D deficiency was reached when the 25(OH)D concentration was <20 ng/dL [28].

A diagnosis of MS was reached in the adolescents according to the criteria of the IDF: the presence of abdominal obesity (≥p90) associated with two or more clinical criteria, such as high BP (systolic/diastolic BP ≥ 130 or ≥85), hyperglycemia (≥100 mg/dL), hypertriglyceridemia (≥150 mg/dL), and low HDL-c levels (<40 mg/dL) [26]. 

### 2.3. Genetics Analyses

Genomic DNA was obtained from whole blood using the commercial QIAamp DNA Blood Mini Kit (Qiagen, CA, USA) by following the manufacturer’s recommendations. The obtained material was stored in a −20 °C freezer until analysis. The integrity of the DNA samples was evaluated by electrophoretic separation on 0.8% agarose gel in tris borate EDTA (TBE) buffer (pH 8.0), followed by staining with red gel and photodocumentation using a SmartView Pro 1200 Imager System (Major Science, CA, USA). DNA was quantified using the Qubit^®^ 2.0 fluorometer (Foster City, CA, USA).

The search for polymorphisms in the *CYP2R1* and *VDR* was performed with real-time polymerase chain reaction (PCR) in the ABI 7500 Fast device (Applied Biosystems, Foster City, CA, USA) using the allelic discrimination technique (TaqMan^®^ system) and assays provided by Applied Biosystems (Foster City, CA, USA). The following polymorphisms were analyzed: rs10741657 (A > G) (C___2958430_10), rs2060793 (A > G) (C___2958431_10), and rs12794714 (A > G) (C___1131665_10) in the *CYP2R1* and rs2228570 (A > G) (C__12060045_20), rs1544410) (T > C) (C___8716062_20), rs7975232 (A > C) (C__28977635_10), and rs731236 (A > G) (C___2404008_10) in the *VDR*.

### 2.4. Statistical Analysis

Statistical analyses were performed using SPSS version 22.0 (SPSS Inc., Chicago, IL, USA). Data were tested for the normal distribution of continuous variables using the Kolmogorov–Smirnov test. Continuous variables were expressed as mean and standard deviation of the mean or median and interquartile range. Variables with parametric distributions were analyzed using Student’s t test or a one-way analysis of variance (ANOVA), followed by Tukey’s post-test. Variables that were considered non-parametric were analyzed using the Mann–Whitney test or Kruskal–Wallis test, followed by Dunn’s post-test. Differences between categorical variables, sex, genotype/allele frequencies, and Hardy–Weinberg equilibrium were tested using the chi-square (χ^2^) test.

The associations between polymorphisms, vitamin D deficiency, MS, and MS components were evaluated using binary logistic regression, according to the following genetic models (dominant and recessive), which were adjusted for sex and BMI variables [29]. Odds ratios (OR) and 95% CIs were used to assess the strength of the associations. The reference genotype was homozygous for the wild-type allele of each polymorphism. The software used in these assessments was SPSS version 22.0. Linkage disequilibrium (LD) was estimated from the combined data of all groups by calculating r2 [30]. The structure of the haplotype block was determined using the CI algorithm [31], and the haplotype frequencies were estimated using the expectation maximization algorithm [32] in Haploview version 4.1 (Broad Institute, Cambridge, MA, USA). In all the analyses, a *p*-value < 0.05 was considered statistically significant.

## 3. Results

This study included a total of 174 adolescents who were assigned to the non-MS and MS groups. Table 1 shows the characteristics of these subjects. The mean age was similar between the groups (11 ± 10.1 years) (*p* = 0.530), and the frequency of MS was higher in male adolescents (14.9%) than in female adolescents (12.6%). The results showed significantly higher BMI (*p* = 0.003), WC (*p* = 0.017), triglyceride (*p* < 0.001), and fasting glucose (*p* < 0.001) values and significantly lower HDL-c values (*p* < 0.001) in adolescents with MS than in those without MS (Table 1). However, the systolic BP (*p* = 0.072), diastolic BP (*p* = 0.302), low-density lipoprotein cholesterol (LDL-c) (*p* = 0.381), total cholesterol (*p* = 0.980), and 25(OH)D (*p* = 0.114) levels did not differ between the groups.

Hardy–Weinberg equilibrium was verified in the analysis of the *CYP2R1* and *VDR* polymorphisms (Table 2). To ensure scoring quality, 10% of the samples were re-genotyped at random, and all the results were consistent. Genotypes and allele frequencies of the *CYP2R1* and *VDR* polymorphisms in the adolescents in the non-MS and MS groups are shown in Table 2. 

During this evaluation, it was observed that in the case of the single nucleotide polymorphism (SNP), rs12794714, in the *CYP2R1*, the GG genotype (OR = 3.54, 95% CI = 1.24–10.14, *p* = 0.023) and G allele (OR = 1.74, 95% CI = 1.09–2.84, *p* = 0.023) were significantly associated with the risk of MS. In the case of the other *CYP2R1* and *VDR* polymorphisms, no significant associations with the risk of MS development were observed.

Table 3 presents the results of the logistic regression analysis of the association of *CYP2R1* and *VDR* polymorphisms, as well as 25(OH)D levels with MS development. In this evaluation, it was observed that the GG genotype of the *CYP2R1* (SNP rs12794714) polymorphism was associated with the risk of MS (OR = 2.74, 95% CI = 1.14–6.58, *p* = 0.024). No significant associations were observed with the other *CYP2R1* and *VDR* polymorphisms, as well as with the 25(OH)D levels.

To investigate the relationship of each *CYP2R1* (Table 4) and *VDR* (Table 5) polymorphism with MS and vitamin D status, the components of MS and vitamin D deficiency in adolescents with MS were evaluated according to the genetic inheritance model.

Regarding *CYP2R1* (Table 4), it was observed that the GG genotype (recessive model) of the SNP, rs10741657 (OR = 3.90, 95% CI = 1.18–12.92, *p* = 0.026), and AG + GG genotype (dominant model) of the SNP rs2060793 (OR = 0.28, 95% CI = 0.08–0.92, *p* = 0.037) were significantly associated with hyperglycemia protection. No significant associations of the components of MS and vitamin D deficiency with the other *CYP2R1* polymorphisms were observed.

In the evaluation of the *VDR* polymorphisms (Table 5), it was observed that the CC genotype (recessive model) of the SNP rs7975232 was significantly associated with hypertension (OR = 5.91, 95% CI = 1.91–18.32, *p* = 0.002). In the case of the other *VDR* polymorphisms, no significant associations were observed with the MS components and vitamin D deficiency.

The results of the LD analysis of the *CYP2R1* and *VDR* polymorphisms are shown in Figure 1A and 1B, respectively. A single haplotypic block (Figure 1C) was constructed for the SNPs, rs10741657 and rs2060793, in the *CYP2R1*. However, there was no significant association with MS (*p* > 0.05). 

## 4. Discussion

The findings of the present study have revealed an important association between the *CYP2R1* polymorphism, rs12794714, and an increased risk of MS. There are few reports in the literature indicating that *CYP2R1* is associated with metabolic outcomes, such as obesity, hypertension, and diabetes, which are important conditions related to MS. To the best of our knowledge, this is the first time that a variant of the *CYP2R1* has been associated with the risk of MS. The variant rs12794714, which was found to be associated with MS in our study, is located in the promoter region of the *CYP2R1*, which may influence gene transcription, and thus could be considered a possible new molecular marker for predicting the risk of MS [18].

Xu et al. (2022) found that in 766 individuals with incident hypertension, the rs12794714 polymorphism was significantly associated with an increased risk of hypertension (OR = 1.26, 95% CI = 1.01–1.56, *p* = 0.041). Furthermore, interactions between rs12794714 and general (OR = 3.93, CI = 2.72–5.68, *p* < 0.001) and central obesity (OR = 3.22, CI = 2.29–4.52, *p* < 0.001) exerted significant effects on the susceptibility to hypertension in the study population [18]. 

On individually evaluating the MS components, we identified an association between SNP rs10741657 (recessive model) of the *CYP2R1* and hyperglycemia. Furthermore, in the group of individuals with hyperglycemia in our study population, the frequency of the presence of the GG genotype of the SNP rs107441657 (32.3%) was higher than that of the presence of the AA genotype (22.6%). Our findings have revealed, for the first time, that new variants of the *CYP2R1*, which have not yet been reported in the literature, are associated with hyperglycemia. A possible explanation for the association between *CYP2R1* variant and hyperglycemia, which was identified in our study, is that the changes in vitamin D levels caused by these polymorphisms may influence the extracellular calcium concentrations in pancreatic β cells, which in turn may affect calcium-dependent insulin secretion [33]. On the other hand, SNP rs2060793 (dominant model) of the *CYP2R1* was associated with protection from hyperglycemia.

Other SNPs of the *CYP2R1* have been reported to increase the risk of diabetes. Wang et al. (2018) found that the rs1993116 and rs10766197 polymorphisms were significantly associated with the risk of type 2 *Diabetes mellitus*. Carriers of the AG + GG genotype of the rs1993116 and rs10766197 polymorphisms had a higher risk of developing type 2 *Diabetes mellitus* than AA carriers. However, they found no association between the variants, rs10741657 and rs12794714, and type 2 *Diabetes mellitus* [34]. 

Consistent with the results of previous studies, our results demonstrated an association between a *VDR* variant and an almost 6-fold increase in the risk of hypertension [35,36]. The role of vitamin D in the risk of hypertension has been demonstrated in experimental studies, which have indicated that vitamin D may be an endocrine negative regulator of the renin-angiotensin-aldosterone system (RAAS), a key stabilizer of BP balance. This downregulation of 1,25-dihydroxyvitamin D3-mediated renin expression and RAAS activity occurs through its interaction with the vitamin D receptor. Global *VDR*-knockout mice have been reported to have higher BP and to develop cardiac hypertrophy due to increased renin expression and subsequent RAAS activation [37,38,39].

However, few studies were able to reproduce a direct relationship between hypertension and rs7975232 (*Apa*I). This is an important variant that is located in the three prime untranslated (3’-UTR) region of the *VDR* and may influence messenger ribonucleic acid (mRNA) stability and VDR protein expression. Hajj et al. (2016) observed an association between *ApaI* and BP in a population of 369 adults. Women with the TT genotype had higher BP values [40]. However, the role of *VDR* variants in BP remains controversial. A large genetic study failed to reproduce any association between *VDR*-related SNPs and BP, suggesting that further research is needed to elucidate this association [41].

Our study population consisted solely of adolescents with overweight. It is known that obesity exerts an influence on vitamin D concentrations, thereby hindering its biological role within cells, and that vitamin D deficiency can influence weight gain and other metabolic changes, which are well established in the literature [5,10]. Thus, the influence of genetics on these outcomes has been increasingly investigated in order to identify new molecular markers that can predict the risk of disease in vulnerable populations. There is evidence of a relationship of both the genes evaluated in our study with the risk of obesity in other populations. Although we did not identify an association between polymorphisms and obesity or vitamin D deficiency itself in our study, it was possible to identify an association between these variants and important risk factors for MS. These findings reinforce the importance of evaluating populations globally, as vitamin D deficiency, hyperglycemia, hypertension, and MS are increasingly being investigated because these conditions are directly linked to obesity; further, when these conditions are present in pediatric populations, they increase the risk of morbidity and mortality in adulthood.

Our study has some limitations. First, the sample size was relatively small for studies on genetic polymorphisms, which may have influenced the fact that we did not find an association between the studied variants and obesity. Further comprehensive research is required to validate our results. Furthermore, the stage of sexual maturation was not included while analyzing the results; this stage may influence some metabolic parameters at the start of puberty. Finally, not all the *VDR* and *CYP2R1* polymorphisms were included in the assessment of the study population. Therefore, it was not possible to fully assess the influence of SNPs on MS and its components. 

Finally, our study has some strengths. First, our sample consisted entirely of adolescents with overweight/obesity; however, the lack of data on comorbidities and use of medications could have affected the analysis of our results. Second, the seven selected variants were previously published in genome-wide association studies (GWAS) that evaluated their biological effects. Third, all the patients were selected from a referral endocrinology center in the state of Rio de Grande do Norte, reinforcing the representative nature of the sample. Finally, our results reinforce the importance of evaluating molecular markers and anthropometric and biochemical variables together in order to predict the risk of MS.

## 5. Conclusions

The present study demonstrated that the rs12794714 polymorphism in the *CYP2R1* was significantly associated with an increased risk of MS, regardless of 25(OH)D concentrations; this finding revealed its potential as a novel molecular biomarker for predicting the risk of MS in our population. Furthermore, we observed that both the rs10741657 and rs2060793 polymorphisms of the *CYP2R1* can significantly influence glucose metabolism, while the rs7975232 polymorphism of the *VDR* was significantly associated with the risk of hypertension. 

These results reinforce the importance of looking for genetic markers associated with vitamin D metabolism in overweight/obese children and adolescents in order to identify, at an early stage, variants associated with MS components, since morbidity and mortality progress more rapidly when starts in young individuals, with unfavorable prognostic implications and a potential increase in individual and collective costs.

## Figures and Tables

**Figure 1 nutrients-14-04612-f001:**
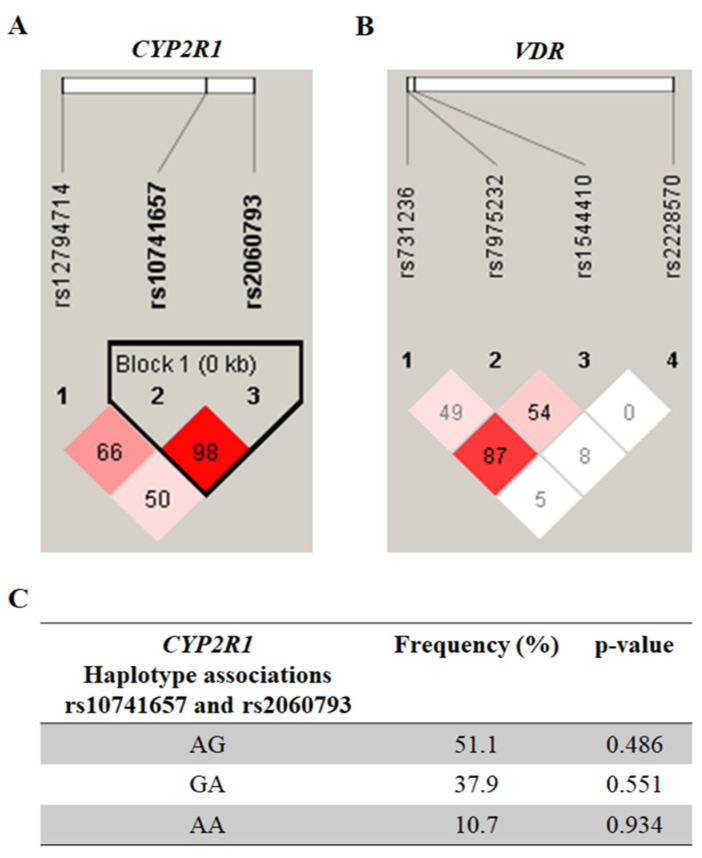
Pairwise linkage disequilibrium plot of *CYP2R1*/rs12794714, rs10741657, and rs2060793 (**A**), as well as of *VDR*/rs731236, rs7975232, rs1544410, and rs2228570 (**B**) in the combined sample (cases & controls) showing r2 (×100) values. The block for *CYP2R1* (**C**) is designed according to the internally developed solid spine of linkage disequilibrium (LD). The value within each diamond represents the pairwise correlation between pairs of single-nucleotide polymorphisms (SNPs) (measured as 100 × r2) defined by the upper left and upper right sides of the diamond. The frequency of each haplotype is shown below the blocks.

**Table 1 nutrients-14-04612-t001:** Characteristics of the adolescents in the non-MS and MS groups.

Variables	Non-MS (*n* = 126)	MS (*n* = 48)	*p*-Value ^a^
Age, mean (SD), years	11 (10.1)	11 (10.1)	0.530
Sex (female n (%)/male n (%)	61 (48.4)/65 (51.6)	22 (45.8)/26 (54.2)	0.761
BMI, mean (SD), kg/m^2^	26.5 (3.7)	28.4 (4)	**0.003**
WC, mean (SD), cm	86 (10)	91 (17)	**0.017**
SBP, median (IQR), mmHg	110 (109–120)	120 (110–130)	0.072
DBP, median (IQR), mmHg	70 (65–80)	70 (60–80)	0.302
Fasting glucose, mean (SD), mg/dL	92 (7)	98 (8)	**<0.001**
Total cholesterol, mean (SD), mg/dL	172 (31)	171 (36)	0.980
HDL-c, mean (SD), mg/dL	42 (7)	33 (5)	**<0.001**
LDL-c, mean (SD), mg/dL	110 (26)	106 (27)	0.381
Triglycerides, median (IQR), mg/dL	93 (71–119)	158 (108–209)	**<0.001**
25-hydroxyvitamin D, mean (SD), ng/dL	31.9 (10.3)	29.9 (8.2)	0.114
25-hydroxyvitamin D deficiency, n (%)	9 (5.2)	8 (4.7)	0.082

Data are presented as mean ± standard deviation of the mean or median (interquartile range), unless otherwise indicated. ^a^
*p*-value obtained by Student’s *t* test, Mann-Whitney test or χ^2^ test. Significant *p*-values are shown in bold. BMI, body mass index; DBP, diastolic blood pressure; HDL-c, high-density lipoprotein cholesterol; IQR, interquartile range; LDL-c, low-density lipoprotein cholesterol; MS, metabolic syndrome; *n*, number of individuals; SD, standard deviation; SBP, systolic blood pressure; WC, waist circumference.

**Table 2 nutrients-14-04612-t002:** Genotype and allele frequencies of the *CYP2R1* and *VDR* polymorphisms in the adolescents in the non-MS and MS groups.

SNPs	Genotypes /Alleles	Non-MS (*n* = 126)	MS (*n* = 48)	^a^*p*-Value	OR	95% CI	^b^*p*-Value
*CYP2R1*							
rs10741657 (A > G)	AA	43 (34.1)	20 (41.7)	0.859	1.00	-	-
	AG	67 (53.2)	22 (45.8)		0.71	0.34–1.45	0.362
	GG	16 (12.7)	6 (12.5)		0.81	0.27–2.37	0.792
	A	153 (60.7)	62 (64.6)		1.00	-	-
	G	99 (39.3)	34 (35.4)		0.84	0.52–1.37	0.538
rs2060793 (A > G)	AA	17 (13.5)	11 (22.9)	0.999	1.00	-	-
	AG	93 (73.8)	30 (62.5)		0.50	0.21–1.18	0.156
	GG	16 (12.7)	7 (14.6)		0.68	0.21–2.17	0.567
	A	127 (50.4)	52 (54.2)		1.00	-	-
	G	125 (49.6)	44 (45.8)		0.86	0.54–1.36	0.550
rs12794714 (A > G)	AA	31 (24.6)	7 (14.6)	0.149	1.00	-	-
	AG	75 (59.5)	25 (52.1)		1.48	0.58–3.77	0.502
	GG	20 (15.9)	16 (33.3)		3.54	1.24–10.14	**0.023**
	A	137 (54.4)	39 (40.6)		1.00	-	-
	G	115 (45.6)	57 (59.4)		1.74	1.09–2.84	**0.023**
*VDR*							
rs2228570 (A > G)	AA	57 (45.2)	20 (41.7)	0.587	1.00	-	-
	AG	51 (40.5)	20 (41.7)		1.12	0.54–2.31	0.853
	GG	18 (14.3)	8 (16.7)		1.27	0.48–3.36	0.620
	A	165 (65.5)	60 (62.5)		1.00	-	-
	G	87 (34.5)	36 (37.5)		1.14	0.70–1.85	0.617
rs731236 (A > G)	AA	28 (22.2)	15 (31.2)	0.435	1.00	-	-
	AG	77 (61.1)	27 (56.2)		0.65	0.3–1.41	0.317
	GG	21 (16.7)	6 (12.5)		0.53	0.18–1.61	0.296
	A	133 (52.8)	57 (59.4)		1.00	-	-
	G	119 (47.2)	39 (40.6)		0.76	0.47–1.22	0.280
rs1544410 (T > C)	TT	30 (23.8)	12 (25.0)	0.061	1.00	-	-
	TC	82 (65.1)	31 (64.6)		0.95	0.43–2.08	0.999
	CC	14 (11.1)	5 (10.4)		0.89	0.26–3.03	0.998
	T	142 (56.3)	55 (57.3)		1.00	-	-
	C	110 (43.7)	41 (42.7)		0.96	0.59–1.53	0.617
rs7975232 (A > C)	AA	45 (35.7)	15 (31.2)	0.217	1.00	-	-
	AC	51 (40.5)	19 (39.6)		1.12	0.51–2.45	0.842
	CC	30 (23.8)	14 (29.2)		1.40	0.59–3.32	0.509
	A	141 (56)	49 (51)		1.00	-	-
	C	111 (44)	47 (49)		1.22	0.76–1.94	0.470

Data are presented as absolute and relative frequencies. ^a^ *p*-value for the Hardy–Weinberg equilibrium test. ^b^ *p*-value for OR. Differences between genotype/allele frequencies and Hardy–Weinberg equilibrium were tested by χ^2^ test. Significant *p*-values are shown in bold. CI, confidence interval; MS, metabolic syndrome; OR, odds ratio; SNP, single-nucleotide polymorphism.

**Table 3 nutrients-14-04612-t003:** Logistic regression analysis of the association of the *CYP2R1* and *VDR* polymorphisms and 25(OH)D levels with MS development.

Variables	OR (95% CI)	*p*-Value
*CYP2R1* genotypes (SNP)		
GG (rs10741657)	1.52 (0.52–4.43)	0.443
GG (rs2060793)	1.45 (0.57–3.68)	0.438
GG (rs12794714)	2.74 (1.14–6.58)	**0.024**
*VDR* genotypes (SNP)		
GG (rs2228570)	0.74 (0.27–2.04)	0.561
GG (rs731236)	0.85 (0.23–3.18)	0.807
CC (rs1544410)	1.25 (0.29–5.43)	0.770
CC (rs7975232)	1.67 (0.74–3.76)	0.218
25-hydroxyvitamin D	0.97 (0.93–1.01)	0.134

Logistic regression analysis with MS as dependent and SNPs of the *CYP2R1* (rs10741657, rs2060793 and rs12794714) and *VDR* (rs2228570, rs731236, rs1544410 and rs7975232), as well as, 25-hydroxyvitamin D levels as covariates. Model adjusted for sex and BMI. Significant *p*-values are shown in bold. CI, confidence interval; OR, odds ratio; SNP, single-nucleotide polymorphism.

**Table 4 nutrients-14-04612-t004:** Association of *CYP2R1* polymorphisms with components of MS and vitamin D deficiency.

SNPs/ Models	Genotypes	Abdominal Obesity		Hyperglycemia		Hypertension		Low HDL-c		High TG		VitD Deficiency	
		OR (95% CI)	*p*-Value	OR (95% CI)	*p*-Value	OR (95% CI)	*p*-Value	OR (95% CI)	*p*-Value	OR (95% CI)	*p*-Value	OR (95% CI)	*p*-Value
rs10741657													
Dominant	AA	1	0.099	1	0.187	1	0.060	1	0.236	1	0.488	1	0.499
	AG + GG	0.97 (0.93–1.01)		1.97 (0.72–5.38)		2.89 (0.96–8.69)		0.64 (0.30–1.34)		1.35 (0.58–3.17)		0.78 (0.38–1.60)	
Recessive	AA + AG	1	0.726	1	**0.026**	1	0.974	1	0.149	1	0.394	1	0.343
	GG	1.01 (0.95–1.07)		3.90 (1.18–12.92)		1.03 (0.23–4.61)		0.43 (0.14–1.35)		1.67 (0.52–5.38)		0.59 (0.20–1.74)	
rs2060793													
Dominant	AA	1	0.738	1	**0.037**	1	0.970	1	0.101	1	0.490	1	0.226
	AG + GG	0.99 (0.93–1.05)		0.28 (0.08–0.92)		1.03 (0.23–4.58)		2.57 (0.83–7.95)		0.67 (0.21–2.12)		1.92 (0.67–5.53)	
Recessive	AA + AG	1	0.830	1	0.076	1	0.430	1	0.593	1	0.358	1	0.162
	GG	1.04 (0.99–1.10)		0.15 (0.02–1.22)		0.59 (0.16–2.17)		1.29 (0.50–3.31)		1.60 (0.59–4.31)		1.97 (0.76–5.11)	
rs12794714													
Dominant	AA	1	0.406	1	0.415	1	0.409	1	0.573	1	0.739	1	0.079
	AG + GG	1.02 (0.97–1.07)		0.65 (0.23–1.88)		1.80 (0.45–7.21)		1.28 (0.54–3.02)		0.85 (0.33–2.19)		2.11 (0.32–4.83)	
Recessive	AA + AG	1	0.542	1	0.992	1	0.994	1	0.288	1	0.259	1	0.308
	GG	1.01 (0.97–1.06)		1.01 (0.33–3.09)		1.01 (0.31–3.21)		1.62 (0.66–3.97)		1.70 (0.68–4.30)		1.57 (0.66–3.72)	

*p*-values were calculated using a logistic regression analysis. Models adjusted for sex and BMI. Significant *p*-values are shown in bold. CI, confidence interval; HDL-C, high-density lipoprotein cholesterol; OR, odds ratio; SNP, single-nucleotide polymorphism; TG, triglycerides; VitD, vitamin D.

**Table 5 nutrients-14-04612-t005:** Association of *VDR* polymorphisms with components of MS and vitamin D deficiency.

SNPs/ Models	Genotypes	Abdominal Obesity		Hyperglycemia		Hypertension		Low HDL-c		High TG		VitD Deficiency	
		OR (95% CI)	*p*-Value	OR (95% CI)	*p*-Value	OR (95% CI)	*p*-Value	OR (95% CI)	*p*-Value	OR (95% CI)	*p*-Value	OR (95% CI)	*p*-Value
rs2228570													
Dominant	AA	1	0.412	1	0.167	1	0.597	1	0.945	1	0.621	1	0.064
	AG + GG	1.02 (0.98–1.06)		0.52 (0.21–1.31)		1.31 (0.48–3.60)		1.03 (0.50–2.11)		1.23 (0.54–2.78)		0.51 (0.25–1.04)	
Recessive	AA + AG	1	0.074	1	0.634	1	0.598	1	0.532	1	0.161	1	0.485
	GG	1.05 (0.10–1.11)		0.72 (0.18–2.83)		0.69 (0.17–2.77)		0.72 (0.26–2.00)		0.33 (0.07–1.55)		0.70 (0.25–1.92)	
rs731236													
Dominant	AA	1	0.366	1	0.333	1	0.506	1	0.435	1	0.814	1	0.363
	AG + GG	1.02 (0.98–1.07)		0.69 (0.55–5.82)		1.79 (0.23–2.07)		0.72 (0.32–1.64)		0.90 (0.37–2.20)		0.69 (0.31–1.53)	
Recessive	AA + AG	1	0.326	1	0.675	1	0.300	1	0.874	1	0.862	1	0.903
	GG	0.98 (0.93–1.03)		0.75 (0.20–2.88)		1.96 (0.55–6.97)		1.08 (0.41–2.85)		0.91 (0.30–2.71)		1.06 (0.42–2.69)	
rs1544410													
Dominant	TT	1	0.577	1	0.221	1	0.385	1	0.898	1	0.757	1	0.121
	TC + CC	1.01 (0.97–1.06)		2.26 (0.61–8.33)		1.76 (0.49–6.24)		1.06 (0.46–2.42)		0.87 (0.35–2.15)		0.52 (0.23–1.19)	
Recessive	TT + TC	1	0.249	1	0.901	1	0.536	1	0.518	1	0.909	1	0.881
	CC	0.97 (0.91–1.02)		1.09 (0.27–4.37)		1.61 (0.35–7.31)		1.45 (0.47–4.44)		1.07 (0.32–3.62)		0.92 (0.32–2.67)	
rs7975232													
Dominant	AA	1	0.521	1	0.820	1	0.281	1	0.670	1	0.606	1	0.978
	AC + CC	1.01 (0.97–1.05)		0.894 (0.34–2.35)		1.88 (0.60–5.89)		1.18 (0.55–2.50)		0.80 (0.35–1.84)		0.99 (0.48–2.04)	
Recessive	AA + AC	1	0.230	1	0.183	1	**0.002**	1	0.299	1	0.126	1	0.509
	CC	0.97 (0.93–1.02)		2.00 (0.72–5.53)		5.91 (1.91–18.32)		1.59 (0.67–3.78)		0.42 (0.14–1.28)		0.75 (0.33–1.74)	

*p*-values were calculated using a logistic regression analysis. Models adjusted for sex and BMI. Significant *p*-values are shown in bold. CI, confidence interval; HDL-C, high-density lipoprotein cholesterol; OR, odds ratio; SNP, single-nucleotide polymorphism; TG, triglycerides; VitD, vitamin D.

## Data Availability

The data presented in this study are available on request from the corresponding author. The data are not publicly available due to privacy restrictions.

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
