# Peer review of "Association of CYP2R1 and VDR Polymorphisms with Metabolic Syndrome Components in Non-Diabetic Brazilian Adolescents"

_nutrients, 2022, doi:10.3390/nu14214612_

Round 1

Reviewer 1 Report

This manuscript presents a study on the effect of polymorphism of CYP2R1 on the obesity of young patients in Brasil and also on the metabolic syndrome (MS) using strong statistical methods.

It is not clear to me what control population was to calculate the OR value. THis should be better explained.

I also am on the impression that the size of the population used is a little small to study this number of SNPs (about 7). But I am not a specialist of genetic polymorphism.

It would be also nice to explain what reactions steps the CYP2R1 performs on the vitamin D metabolism. This is not clearly stated.

The authors report a new polymorphism associated with MS. This is interesting.

I find some of the tables (with 2 lines per parameter) a little difficult to read. Can you simplify them.

As such the paper presnt some intersting results. The paper could be edited to show more clearly the results obtained.

It should be acceptable with minor editing.

Author Response

Response to Reviewer 1 Comments

Point 1: It is not clear to me what control population was to calculate the OR value. This should be better explained.

Response 1: We appreciate the comment. The study design adopted a total sample number of overweight/obesity individuals and from this sample they were subdivided into two groups, with MS and without MS. The OR calculation was performed in relation to the group that does not have MS. Additional information about the groups studied was inserted in the methodology, on line 106, page 3.

The following sentences in the previous manuscript (lines 103-106):“ The study participants were classified according to the anthropometric nutritional status using the World Health Organization (WHO) BMI/age curves according to sex; z-scores between +1 and +2 indicated overweight, z-scores between +2 and +3 indicated obesity, and z-scores ≥ +3 indicated severe obesity [25]. It was replaced by (Please see lines 103-108, page 3):“ The study participants were classified according to the anthropometric nutritional status using the World Health Organization (WHO) BMI/age curves according to sex; z-scores between +1 and +2 indicated overweight, z-scores between +2 and +3 indicated
obesity, and z-scores ≥ +3 indicated severe obesity [25]. Then, the adolescents were subdivided into groups with MS (n=48) and without MS (n=126), according to the criteria proposed by the International Diabetes Federation (IDF) [26]’’.

Point 2: I also am on the impression that the size of the population used is a little small to study this number of SNPs (about 7). But I am not a specialist of genetic polymorphism.

Response 2: We appreciate the comment. We clarify that in studies of genetic polymorphisms, the sample number is not calculated from the number of polymorphisms evaluated, but in relation to the number of individuals studied for each of the SNPs. Therefore, the sample calculation was performed as described in line 96, page 3, resulting in a number of 174 overweight/obesity adolescents.

 Point 3: It would be also nice to explain what reactions steps the CYP2R1 performs on the vitamin D metabolism. This is not clearly stated.

Response 3: We appreciate the reviewer's suggestion. We have rewritten and more clearly inserted additional information about the steps that CYP2R1 takes in vitamin D metabolism in the introduction (Please see line 55, page 2).

The following sentences in the previous manuscript (lines 54-57, page 2):“Among the main known genes involved in the vitamin D metabolic pathway, the CYP2R1 is responsible for the first hydroxylation of 25-hydroxyvitamin D (25(OH)D) in the liver.” It was replaced by (Please see lines 54-58, page 2):

“Among the main genes involved in the bioactivation of vitamin D, CYP2R1 is responsible for the hydroxylation of vitamin D3 to 25-hydroxyvitamin D3 (25(OH)D). This process initially occurs when the 25-hydroxylase enzyme (encoded by the CYP2R1 gene) converts vitamin D (inactive precursor derived from sunlight exposure or dietary intake) to 25(OH)D (circulating form) in the liver.”

Point 4: I find some of the tables (with 2 lines per parameter) a little difficult to read. Can you simplify them.

Response 4: Thanks to the reviewer for this comment. We adapted the tables 4 and 5 according to the suggestions provided. (Please, see line 244, page 8 and line 255, page 9).

Dear Editor and Reviewer, we thank you again for your kindness in giving us the opportunity to resubmit our manuscript for further review.

Reviewer 2 Report

In this work, the authors investigate the association of CYP2R1 and VDR Polymorphisms with Metabolic Syndrome Components in Non-Diabetic Brazilian Adolescents. The manuscript is well written, I have minor comments and questions as described below.

  1. Have you conducted any analysis based on EHR datasets? Many studies have been able to leverage EHR to validate these associations.   Barnette DA et al. Meloxicam methyl group determines enzyme specificity for thiazole bioactivation compared to sudoxicam. Toxicol Lett. 2021 Mar 1;338:10-20. doi: 10.1016/j.toxlet.2020.11.015. Epub 2020 Nov 27. PMID: 33253783; PMCID: PMC7807415.   There is a growing body of literature in this space that you should cite in your paper that studies how associations lead to increased risks. These are methodological advances and relevant to your work.   Datta A, Flynn NR, Barnette DA, Woeltje KF, Miller GP, Swamidass SJ (2021) Machine learning liver-injuring drug interactions with non-steroidal anti-inflammatory drugs (NSAIDs) from a retrospective electronic health record (EHR) cohort. PLoS Comput Biol 17(7): e1009053. https://doi.org/10.1371/journal.pcbi.1009053   A. Datta et al., "‘Black Box’ to ‘Conversational’ Machine Learning: Ondansetron Reduces Risk of Hospital-Acquired Venous Thromboembolism," in IEEE Journal of Biomedical and Health Informatics, vol. 25, no. 6, pp. 2204-2214, June 2021, doi: 10.1109/JBHI.2020.3033405.   2. What are the potential confounding variables in the study?   3. Your study subject selection is comprehensive. How did you handle thyroid? Also, did you factor in any signals for mobility since low mobility increases risk of obesity? What about food habit patterns?

Author Response

Response to Reviewer 2 Comments

 Point 1: Have you conducted any analysis based on EHR datasets? Many studies have been able to leverage EHR to validate these associations.   Barnette DA et al. Meloxicam methyl group determines enzyme specificity for thiazole bioactivation compared to sudoxicam. Toxicol Lett. 2021 Mar 1;338:10-20. doi: 10.1016/j.toxlet.2020.11.015. Epub 2020 Nov 27. PMID: 33253783; PMCID: PMC7807415.   There is a growing body of literature in this space that you should cite in your paper that studies how associations lead to increased risks. These are methodological advances and relevant to your work.   Datta A, Flynn NR, Barnette DA, Woeltje KF, Miller GP, Swamidass SJ (2021) Machine learning liver-injuring drug interactions with non-steroidal anti-inflammatory drugs (NSAIDs) from a retrospective electronic health record (EHR) cohort. PLoS Comput Biol 17(7): e1009053. https://doi.org/10.1371/journal.pcbi.1009053   A. Datta et al., "‘Black Box’ to ‘Conversational’ Machine Learning: Ondansetron Reduces Risk of Hospital-Acquired Venous Thromboembolism," in IEEE Journal of Biomedical and Health Informatics, vol. 25, no. 6, pp. 2204-2214, June 2021, doi: 10.1109/JBHI.2020.3033405.  

Response 1: We appreciate the suggestion, however, in Brazil, to date, there is no unified electronic system, similar to the EHR, that allows the search and joint analysis of the data evaluated in our study. Therefore, our evaluation was performed using conventional methods of association of the studied polymorphisms with the risk of MS, from inheritance models.

Point 2. What are the potential confounding variables in the study?  

Response 2: We appreciate the observation. Gender and BMI were considered as confounding variables, being used to adjust the regression analyses, according to line 171, page 4.

Point 3: Your study subject selection is comprehensive. How did you handle thyroid? Also, did you factor in any signals for mobility since low mobility increases risk of obesity? What about food habit patterns?

Response 3: We appreciate the comment. We were careful to exclude patients from our sample who had alterations in thyroid function and we clarified the methodology (please see line 109 – 113, page 3). As for information related to mobility and dietary patterns, data were collected, however in previous analyzes it was identified that there were no differences between the groups with MS and without MS.

 The following sentences in the previous manuscript (lines 109-113, page 3):

“The exclusion criteria were the presence of genetic syndromes associated with obesity or other chronic diseases; pregnancy and lactation; use of vitamin D supplementation; use of drugs to treat insulin resistance or type 2 Diabetes mellitus; acute or chronic liver, kidney, or heart failure; cancer; or other conditions that alter vitamin D metabolism”.

It was replaced by (Please see lines 109-113, page 3):

“The exclusion criteria were the presence of genetic syndromes associated with obesity or other chronic diseases; pregnancy and lactation; use of vitamin D supplementation; use of drugs to treat insulin resistance or type 2 Diabetes mellitus; acute or chronic liver, kidney, thyroid dysfunction, heart failure; cancer; or other conditions that alter vitamin D metabolism”.

Dear Editor and Reviewer, we thank you again for your kindness in giving us the opportunity to resubmit our manuscript for further review.

Round 2

Reviewer 2 Report

The authors are publishing in a high impact International journal. Being said that It should n't be country specific data set. I would highly recommend the authors to include some paragraphs based on the first comment of the reviewer in the introduction part of the manuscript citing the important citations as described in the reviewers report. Including those references and machine learning will increase the value of the paper in the research field.